# Digitally Printed AgNPs Doped TiO$_2$ on Commercial Porcelain-Grès Tiles: Synergistic Effects and Continuous Photocatalytic Antibacterial Activity

**Claudia Letizia Bianchi** [1,*], **Giuseppina Cerrato** [2], **Bianca Maria Bresolin** [1,3], **Ridha Djellabi** [4] **and Sami Rtimi** [5]

1 Department of Chemistry, Università degli Studi di Milano, Via Golgi 19, 20133 Milano, Italy; Bianca.Maria.Bresolin@lut.fi
2 Department of Chemistry, Università degli Studi di Torino, Via Giuria 7, 10125 Torino, Italy; giuseppina.cerrato@unito.it
3 Laboratory of Green Chemistry, School of Engineering Science, Lappeenranta University of Technology, Sammonkatu 12, 50130 Mikkeli, Finland
4 Research Center for Eco-Environmental Sciences, Chinese Academy of Sciences, Beijing 100085, China; ridha.djellabi@szu.edu.cn
5 Institute of Materials, Swiss Federal Institute of Technology Lausanne (EPFL), EPFL-STI-IMX-LTP, Station 12, CH-1015 Lausanne, Switzerland; sami.rtimi@epfl.ch
* Correspondence: claudia.bianchi@unimi.it

**Abstract:** In the present study, we use commercial digitally printed ceramic tiles, functionnalized by AgNPs doped micro–TiO$_2$, to investigate the mechanism of Ag in the continouos photocatalytic antibacterial activity. The novelty of the research lies in the attempt to understand the mechanism of Ag, supported on TiO$_2$, able to exhibit the same antibacterial activity of a standard system containing Ag species, but here, totally embedded on the tile surface, and thus not free to move and damage the bacteria cell. UV/vis diffuse reflectance spectroscopy (DRS) of AgNPs–TiO$_2$ tiles indicated an enhanced visible light response, wherein a new absorption band was produced around 18,000–20,000 cm$^{-1}$ (i.e., in the 400–600 nm range) owing to the surface plasmon resonance (SPR) of AgNPs. The antibacterial photocatalytic experiments were conducted towards the inactivation of *E. coli* under solar light and indoor light. It was found that the degradation speed of *E. coli* in the presence of AgNPs–TiO$_2$ tiles is solar light-intensity depending. This justifies the semiconductor behavior of the material. Furthermore, the AgNPs–TiO$_2$ tiles exhibit a high ability for the inactivation of *E. coli* at a high load ($10^4$–$10^7$ colony-forming unit (CFU)/mL). Additionally, AgNPs–TiO$_2$ tiles showed a remarkable antibacterial activity under indoor light, which confirms the good photocatalytic ability of such tiles. On the basis of the reactive oxygen species (ROS) quenching experiments, O$_2$$^{\bullet-}$ species and h$^+$ were more reactive for the inactivation of *E. coli* rather than $^{\bullet}$OH species. This is because of the different lifetime (bacteria are more likely oxidized by ROS with longer lifetime); in fact, O$_2$$^{\bullet-}$ and h$^+$ exhibit a longer lifetime compared with $^{\bullet}$OH species. The generation of H$_2$O$_2$ as the most stable ROS molecule was also suggested.

**Keywords:** continuous antibacterial activity; self-cleaning surface; AgNPs doped TiO$_2$; indoor and outdoor photocatalysis; commercial photocatalytic ceramic tiles

## 1. Introduction

Surface cleaning causes considerable trouble, high consumption of energy, and chemical detergents; consequently, photocatalytic building materials such as cements, tiles, and paints are a current

production as well as scientific reality [1]. It has now been amply demonstrated that they are photo-active and usually self-cleaning materials, as well allowing facade cleaning thanks to sunlight and rain without the need for additional maintenance costs. In the interior, they act as antipollutants even if the advent of the new LED lights has created significant doubt about the actual efficiency of these materials, as their activation requires UV-A lights.

Currently, one of the most interesting photocatalytic materials on the market is the porcelain stoneware, which combines the beauty of a design product with both the hardness and absence of porosity, and thus is used for both floors and walls [2–4].

The micrometric $TiO_2$ decoration with Ag species allowed active ceramic slabs to be active under LED lights, bypassing the problem of the UVA radiation. The ceramic is digitally printed [5] with a tailored ink containing the AgNPs–$TiO_2$, obtaining a final surface with an homogeneous distribution of the photocatalytic micro-sized particles.

Since the age of ancient Egyptians, Ag and Ag-based materials have been extensively used for their strong inhibitory potential and broad-spectrum antibacterial activity [6–8]. Nowadays, they continue to be widely used in many modern applications and in many different fields [9–11].

At macro-scale, the antimicrobial mechanism of Ag is attributed to the release of $Ag^+$ ions able to interact and damage microbe structure. The $Ag^+$ ions can damage the cells until its definitive death in different way [12]. For example, by interacting and eventually entering the cell membrane, $Ag^+$ ions can attach the ions exchange ability of the cell, form a complex with DNA, inactivate the enzymes and inhibit their oxidation, shrink the cytoplasm, decrease the metabolism of bacteria making them non-culturable, or finally inhibiting cell growth by depositing in vacuoles and cell walls. Silver metal, silver acetate, silver nitrate, silver protein, and silver sulfadiazine are usually used as sources of Ag species [6]. In contrast to antibiotics, the bacterial resistance against $Ag^+$ species has been observed only rarely and constitutes almost no significant complication. $Ag^+$ species was proven to inhibit bacterial growth already at low concentrations with significantly low cytotoxicity [13], on the other hand, overdose usage can lead to some complications such as argyria [13].

In the last decades, nanotechnology has been globally considered a novel essence in technological field providing several advantages, especially for its potential of adjustment of the physico-chemical properties of materials [14]. Specifically, Ag species in the form of nanoparticles (AgNPs) were found to be a strategic antibacterial agent mainly owing to their specific surface area that can largely exceed their size. This property was found to remarkably enhance the antibacterial efficiency with respect to the previously described $Ag^+$ species. In contrast, as far as the bactericidal effects of $Ag^+$ species is concerned, colloidal silver particles are strictly influenced by their dimensions; the smaller the particles, the higher the antimicrobial efficiency [15,16], in particular, because both kinetics and bacterial inactivation mechanism are strongly related to the Ag size, shape, and surface charge of the Ag-NPs. Thus, AgNPs have been proposed as a valuable alternative for their antibacterial activity on infections [16,17]. AgNPs can affect microbial growth by damaging the cells until their eventual death with different methodologies. Ag-oxides can catalyze the production of reactive oxygen species (ROS) by affecting the respiration process; they can bid on the cell membrane, leading to a change in membrane properties, and damage its vital functions. Moreover, they can interact with DNA, inhibiting the absorption of phosphates, or by collapsing the proton motive forces. All the previously mentioned antibacterial mechanisms are based on the interaction of the unblocked ability of Ag ions and AgNPs to penetrate and thus interfere at the DNA level in the cellular barrier. Recently, porinless *E. coli* were used under band-gap irradiation in order to differentiate the semiconductor behaviour and the ions mediating the bacterial inactivation mechanism [17].

In our research, we applied a surface modification on commonly used photocatalytic building materials, introducing Ag to induce the antibacterial properties. Even if the concept of Ag-doped $TiO_2$ has been already described in many studies, herein, visible-light-responsive Ag-$TiO_2$ coated tiles were produced in high quantity through an industrial procedure. More importantly, previous studies have ascribed the principal role of the antibacterial response of Ag-$TiO_2$ to the direct attack of the $Ag^+$

ions and AgNPs to the cell wall and core. Notably, in this study, we investigate a new antibacterial mechanism of the as-prepared material. In fact, the core novelty of our study is the development of an antibacterial material characterized by AgNPs blocked on the semiconductor surface. XPS analysis showed the co-existence of both $Ag^+$ and metallic Ag. In particular, the as prepared porcelain Ag-TiO$_2$ printed surfaces were tested under solar and visible light irradiation for antimicrobial applications on Gram-negative pathogenic bacteria such as *E. coli*. Finally, stereomicroscopy analysis showed dead bacteria within 180 min. With appropriate and specially chosen quenchers, the bacterial inactivation mechanisms under solar and visible light are proposed.

## 2. Materials and Methods

### 2.1. Materials

The commercial 1077-Kronos was used as a photoactive semiconductor to design the photoactive tiles. It is pure anatase phase and the main features of this well-known material are summarized in Table 1.

**Table 1.** Main properties of TiO$_2$ (1077, KRONOS Worldwide, Inc., USA).

| BET Surface Area (m$^2$ g$^{-1}$) | Particle Size Range (nm) | XPS Ti 2p$_{3/2}$ (eV) | XPS OH/O$_{tot}$ | Band Gap (eV) |
|---|---|---|---|---|
| 12 ± 2 m$^2$ g$^{-1}$ | 110–130 | 458.4 | 0.32 | 3.15 |

Silver nitrate (AgNO$_3$, ACS Reagent, ≥99%) was used as metal precursor. The other reagents were polyvinylpyrrolidone (PVP40, average mol wt$^{-1}$: 40,000) and sodium borohydride (NaBH$_4$, ≥99%), all Sigma Aldrich reagents used without further purification or pre-treatments.

### 2.2. Synthesis of Anti-Bacterial Ag@TiO$_2$ Photoactive Porcelain-Grès Tiles

The detailed description of the synthesis of Ag@TiO$_2$ photoactive tiles was reported in our previous studies [18,19]. Three different samples with 1%, 4%, and 8% wt AgNPs on micro-sized TiO$_2$ were prepared. Typically, the surface of commercial photoactive porcelain-grès tiles is covered with a formulation containing pure anatase micro-TiO$_2$ (1077 by Kronos) doped with 8% AgNPs from a silver nitrate solution. The digital printing production is applied for the preparation of the functionalized industrial porcelain-grès tiles (Figure 1a). A digital printer (Projecta S.p.A., Italy) impressed the solvent-based ink containing the photocatalyst on tiles surface. After the ink deposition, tiles were calcined at 680 °C for 80 min in a kiln, with high-controlled temperature ramp avoiding thermal shock, and washed with water with a brushing system to remove the particle of non-adherent TiO$_2$. The water used for degreasing was recovered and purified prior to recycling and any damaged tiles were recycled [5]. The main advantages of this procedure are reported as follows: green production with no waste of photocatalytic powder (TiO$_2$), no use of water, smaller consumption of energy, CO$_2$ emissions lower than 89%, possibility to work on large slabs (300 cm × 150 cm).

A remarkable great stability of the photocatalytic coating was achieved. To quickly test the stability of the washcoat adherence, the coated tile was immersed in a vessel containing liquid nitrogen; no detachment of photocatalytic material was observed despite the drastic conditions (Figure 1b).

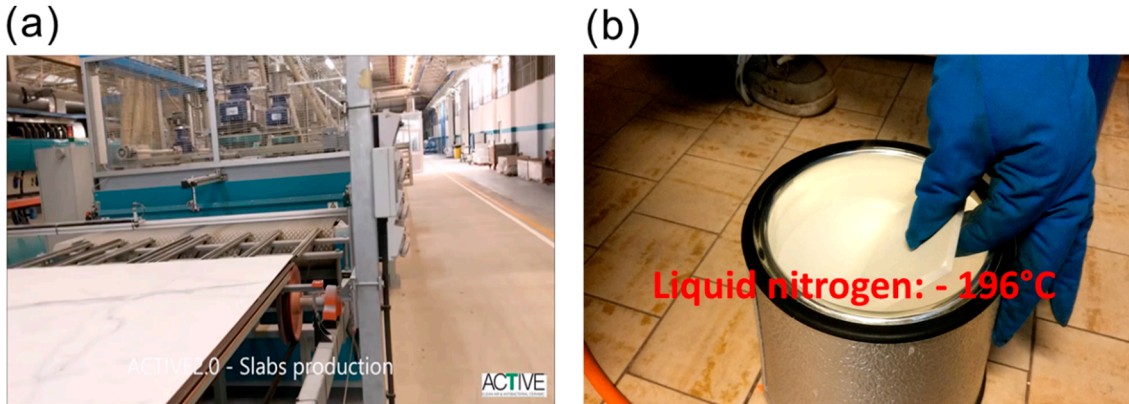

**Figure 1.** (**a**) Synthesis of commercial Ag@TiO$_2$ photoactive porcelain-grès tiles via digital printing production. (**b**) Stability of the Ag@TiO$_2$ photoactive porcelain-grès tiles.

### 2.3. Characteristics of Ag@TiO$_2$ Photoactive Porcelain-Grès Tiles

The material was characterized by means of many different experimental techniques, including HR-SEM, HR-TEM, EDX, and XRD. The detailed description of the performed analyses is reported elsewhere [18]. However, from the EDX analysis, it was possible to determine the amount of TiO$_2$ coated on the tile surface as 1.1 g m$^{-2}$. SEM showed a good distribution of the micro-TiO$_2$ on the porcelain tile surface prepared by the digital printing process.

More details and discussion regarding the characterization of AgNPs–TiO$_2$ photoactive porcelain-grès tiles are reported in our previous research [18,19]. UV/visible diffuse reflectance spectra (UV-DRS) absorption responses of AgNPs–TiO$_2$ samples with different amount of AgNPs loading were obtained by means of a UV/vis scanning spectrophotometer (PerkinElmer, Lambda 35 model, Waltham, MA, USA), equipped with a diffuse reflectance accessory. The possible co-existence of Ag$^+$ and metallic Ag was detected by XPS analysis (M-Probe, SSI Instrument, USA).

### 2.4. Antibacterial Experiments

#### 2.4.1. Photocatalytic Properties

Two different light sources were used during the course of this study, indoor actinic light and solar simulated light. The indoor light irradiation of the samples was carried out in a cavity provided with tubular Osram Lumilux 18W/827, Munich, Germany. These lamps emitted in the range of 340–720 nm with an integral output of 5.0 mW/cm$^2$. A solar simulator for Hereaus GmbH (Hanau, Germany) was used to irradiate the samples with three different irradiances (100, 50, and 20 mW/cm$^2$). The solar simulator emits from a 1500 W Xenon lamp at wavelengths between 310 and 750 nm; a quartz glass cut-off filter cuts the other wavelengths (UVB, UVC, and IR).

The light intensity was monitored using a UV radiometer and a global irradiance couple (CUV3 and CM3 Models, Kipp & Zonen B.V., Delft, The Netherlands).

#### 2.4.2. Bacterial Inactivation and Used Light Sources

The samples of *Escherichia coli* (*E. coli*) were obtained from the Deutsche Sammlung von Mikroorganismen und Zellkulturen GmbH (DSMZ) ATCC23716, Braunschweig, Germany, to test the antibacterial activity of the Ag-coated tiles. Prior to testing, all tiles were autoclaved at 121 °C for 2 h. The bacterial aliquots in NaCl/KCl (pH 7.2) were placed on coated and uncoated (control) tiles. Because the tiles present a micro-porosity, an adsorption stage of 30 min was necessary in order to guarantee an optimal contact between the bacterial cells and the reactive interface. A 100 μL *E. coli* inoculum was contacted with each sample. The inoculation was done at room temperature (25–28 °C). This temperature did not vary much during the bacterial testing. The tiles/samples were then placed in Petri dishes provided with a lid to prevent evaporation, and then exposed to light for the photocatalytic

activity testing. After each determination, the tiles were transferred into a sterile tube containing 5 mL autoclaved saline solution. This solution was subsequently mixed thoroughly using a vortex for 2 min. Serial dilutions were made in NaCl/KCl solution. A 100 µL sample of each dilution was pipetted onto a nutrient agar plate and then spread over the surface of the plate using the standard plate method. These agar plates were then incubated lid down at 37 °C for 24 h before colonies were counted. Three independent assays were done for each sputtered sample. The statistical analysis of the results was performed for the CFU values, calculating the standard deviation values ($n = 5\%$).

To verify that no re-growth of *E. coli* occurs after the first bacterial inactivation cycle, the tiles were incubated for 24 h at 37 °C. These samples were incubated at 37 °C for 24 h. Bacterial regrowth was observed only when using indoor light.

### 2.4.3. Stereomicroscopy and Live/Dead Bacteria

The stereomicroscopy imaging was carried out on samples inoculated with $6 \times 10^6$ CFU/mL of *E. coli* and incubated for 2 h in a humidification chamber. This method uses a fluorochrome ® Biofilm Viability Kit (Molecular Probes, Invitrogen, ThermoFisher Instruments, Waltham, Massachusetts USA). The kit contains a combination of the SYTO9® green fluorescent nucleic acid stain and propidium iodide (PI) fluorochromes for the staining of live and dead cells, respectively. The sample fluorescence was monitored in a stereomicroscope (Leica MZ16 FA, Leica Microsystems GmbH Wetzlar, Germany) and the images were processed using the Leica Application Suite LAS v.1.7.0 build 1240 software from Leica Microsystems CMS GmbH (Wertzlar, Germany). Adhesion of bacteria on the tiles was allowed for 15 min before washing the sample with sterile Milli-Q water to remove non-adherent bacteria.

### 2.4.4. Micro-Oxidation (Local pH) and Interfacial Potential at the Surface of the Coated Tiles

The local pH and interfacial potentials were monitored during the time of bacterial inactivation by means of a Jenco 6230N (pH/mV/temperature meter) with a hand-held microprocessor in splash proof case with a three-point calibration. The signals were monitored via RS-232-C IBM compatible communication interface and BNC, pH/ORP, and eight-pin DIN ATC connector. The redox/pH electrode was standardized using two redox solutions (240 mV and 470 mV; Hanna Instruments, Switzerland) at three pH standards (3, 8, and 10). After each analysis, the electrodes were sterilized in a mixture (70% *v/v*) of distilled water and ethyl alcohol (96% *v/v*) for 16 h. Before each experiment, the electrodes were washed with sterilized MilliQ-water. To maintain the electrode performance, the electrodes were polished after each run with a fine alumina oxide powder (Sigma Aldrich, Buchs Switzerland).

### 2.4.5. ROS Quenchers

Aliquots of the bacterial suspension were suspended on the ceramic tiles in either the absence (control) or presence of chelating agents (ROS quenchers/scavengers), adding a 50 µL of bacterial suspension at a high concentration ($10^6$ CFU/mL). Ethylenediamine-tetra-acetic acid sodium (EDTA-2Na) was added as hole ($h^+$) scavenger. Dimethylsulfoxide (DMSO) was used as ($^\bullet$OH) scavenger and superoxide dismutase (SOD) as $O_2^{\bullet-}$ scavenger. A concentration of 0.2 mM of scavenger was added in each case to identify each intermediate/radical species.

## 3. Results and Discussion

### *3.1. Photocatalyst Characterization*

The morphology of Ag@TiO$_2$ deposited into porcelain-grès tiles was characterized by SEM analysis, as shown in Figure 2a, wherein small sized nanoparticles of Ag@TiO$_2$ were aggregated on the surface, showing a uniform coating. For further clarification, HR-TEM analysis was recorded on bare-micro TiO$_2$ and Ag@TiO$_2$ (Figure 2b,c). Bare TiO$_2$ exhibits a highly ordered structure and the inspection of the fringe patterns confirms the presence of the anatase phase (mostly exposed family planes (101) of anatase [ICDD card n. 21-1272]). In Ag@TiO$_2$ samples, AgNPs are clearly evident on

the surface of TiO$_2$, exhibiting different particles' dimensions in the 3–20 nm range. Both the presence of anatase as unique TiO$_2$ polymorph, for either the bare support and/or all Ag-promoted materials, and the presence of Ag itself have been confirmed by XRD measurements; no figure has been reported for the sake of brevity, as data have been already published elsewhere [20].

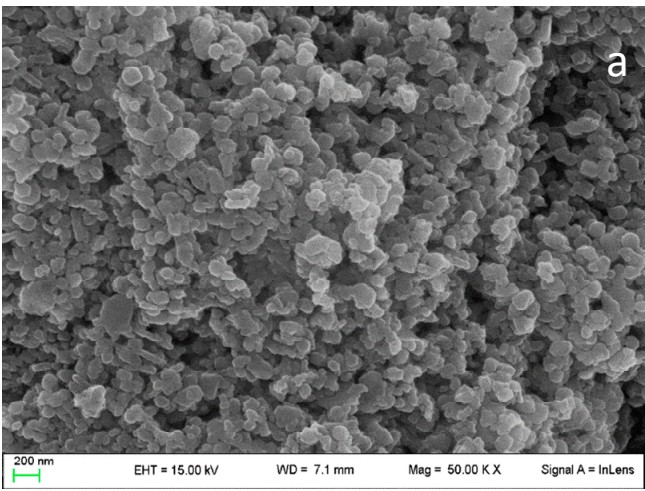

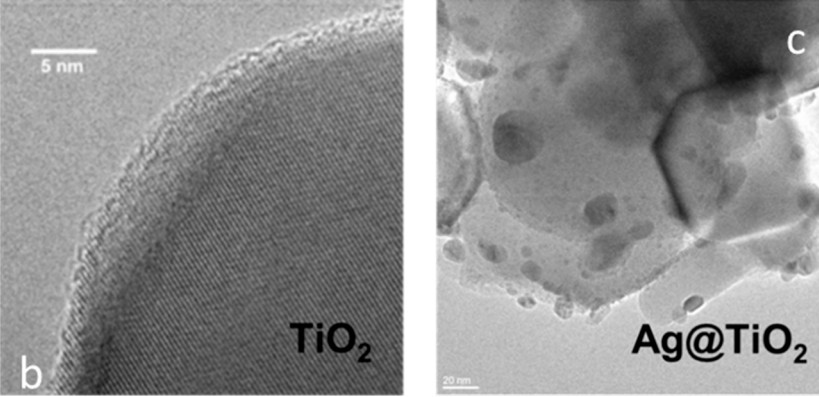

**Figure 2.** (**a**) HR-SEM image of Ag@TiO$_2$ coated onto a porcelain-grès tile; (**b**) HR-TEM image of bare TiO$_2$; and (**c**) HR-TEM image of Ag@TiO$_2$ samples.

In band gap engineering, the doping of TiO$_2$ by cation or anions species is a great approach for enhanced visible light response (bathochromic shift), via two possible mechanistic pathways [21] including either the minimizing of the band gap or the formation of intra-band gap states above the valence band. AgNPs can act as electron receivers and catalytic redox sites in an Ag-doped micro-TiO$_2$ system. Photo-excited electrons of micro-TiO$_2$ are likely transferred from the conduction band of TiO$_2$ to AgNPs because the Fermi levels of Ag are lower than that of TiO$_2$, while the lifetime of oxidative positive holes became longer [22]. This mechanism leads to inhibition of the recombination of electron/hole charges, which in turn enhance the photocatalytic activity. Above all, however, the photocatalytic activity of TiO$_2$ under visible light is significantly boosted when it is assisted by surface plasmon resonance (SPR) [23], wherein the photoexcited electrons under visible light of AgNPs are injected to the conduction band (CB) of TiO$_2$.

UV/visible absorption spectra of AgNPs–TiO$_2$ samples (Figure 3) indicate that the increase of AgNPs amount in TiO$_2$ has almost no effect on the absorption edge of TiO$_2$. On the contrary, the band relative to the Ag plasmon moves towards lower wavenumbers (from 20,600 cm$^{-1}$ for Ag1% to 18,500 cm$^{-1}$ for Ag8%) as the Ag content increases; moreover, the intensity of the plasmon band grows up in intensity as the Ag content increases as well as the average size of the Ag particles. Localized SPR

of AgNPs can significantly enhance the overall conductivity of TiO$_2$ and decrease the recombination of electron/hole charges. Furthermore, a sensitizing Ag$_2$O with TiO$_2$ under visible light may take place. The introduction of AgNPs to TiO$_2$ results in the formation of the Schottky barrier between AgNPs and TiO$_2$ [24]. The SPR photogenerated electrons under visible light can be injected to the conduction band via this barrier, leading to activating TiO$_2$ under visible light [25].

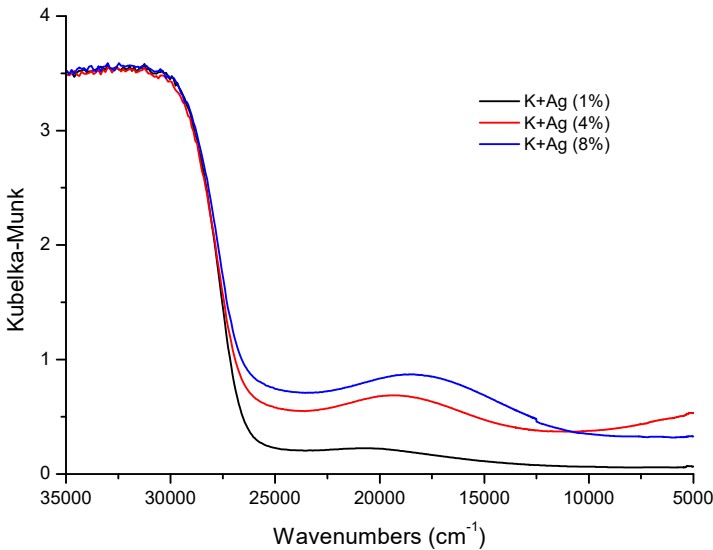

**Figure 3.** UV-diffuse reflectance spectroscopy (DRS) spectra of as-prepared AgNPs–TiO$_2$.

Figure 4 shows the high-resolution XPS spectrum of the Ag 4f$_{7/2}$ and 4f$_{5/2}$ region. The co-existence of Ag$^+$ (Ag$_2$O) and metallic Ag(0) was detected in AgNPs–TiO$_2$ photoactive porcelain-grès tiles [26]. However, it can be noticed that Ag$_2$O is the dominant species compared with metallic Ag.

The main peak at 377 eV is the result of the presence of potassium (K 2s peak), element present as contaminant in the composition of the Kronos K1077 (www.XPSdata.com).

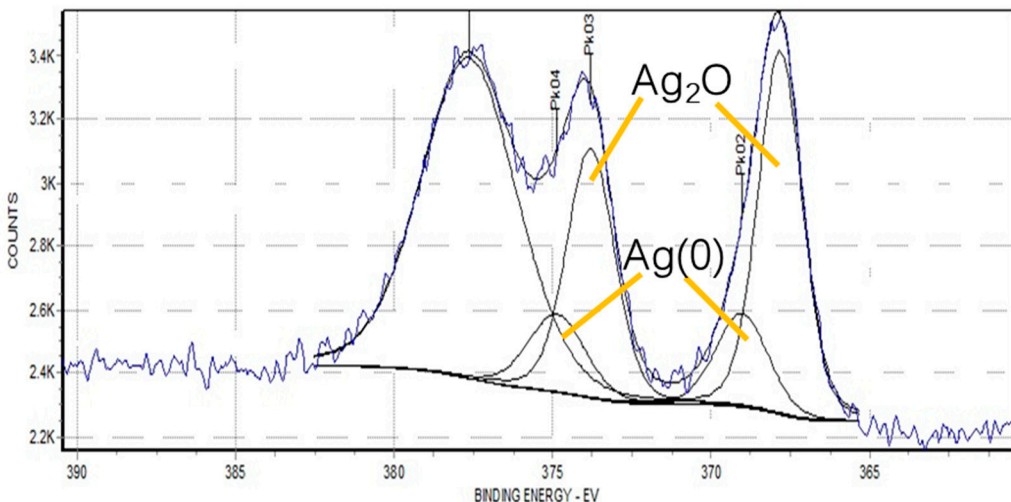

**Figure 4.** High-resolution XPS spectrum of Ag species in AgNPs–TiO$_2$ photoactive porcelain-grès tiles.

### 3.2. E. coli Inactivation under Different Solar Light Intensities

In order to demonstrate the photocatalytic inactivation of *E. coli* on the surface of Ag-NPs–TiO$_2$ tiles, (i) dark run, (ii) photolysis, and (iii) photocatalytic experiments in the presence of Ag-NPs tiles under different light intensities were performed, and the results are shown in Figure 5. It is readily seen that no significant adsorption of *E. coli* on AgNPs–TiO$_2$ tiles was found. Furthermore, the photolysis

of *E. coli* without AgNPs–TiO$_2$ tiles was not obvious within 300 min. However, in the presence of AgNPs–TiO$_2$ tiles under light, a significant inactivation of *E. coli* that increases as a function of light intensity increases. This justifies the semiconductor behaviour of the Ag-species at the interface with bacteria [16,27,28]. Thus, these results indicate that the inactivation of *E. coli* is a light-dependent reaction owing to the formation of ROS on the surface of AgNPs–TiO$_2$.

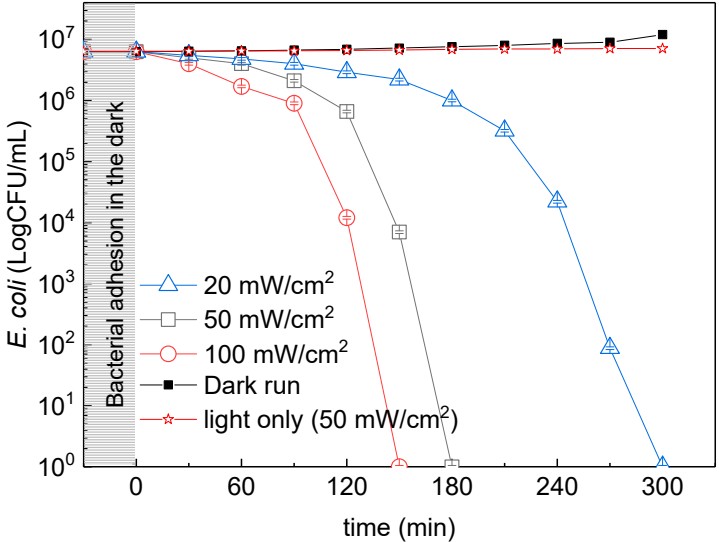

**Figure 5.** *E. coli* inactivation on ceramic tiles coated with 8% Ag-NPs–TiO$_2$ under different solar light intensities.

### 3.3. Different Initial Concentration of Bacteria

Photocatalytic experiments varying *E. coli* initial concentrations were conducted under low intensity solar simulated light (50 mW/cm$^2$) to further check the photocatalytic ability of Ag-NPs–TiO$_2$ tiles (Figure 6). It was found that, even at high concentrations of bacteria, the AgNPs–TiO$_2$ ceramics are able to eradicate bacteria, hindering biofilm formation. In normal settings (hospital rooms, swimming pools, and so on), the concentration of bacteria is in the range of 10$^2$–10$^4$ CFU/mL. The tiles inactivated these concentrations within only 60 min under low intensity solar light.

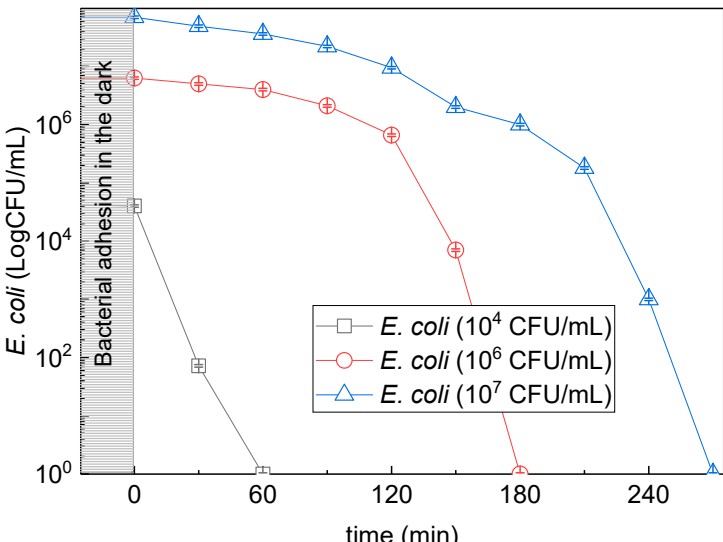

**Figure 6.** *E. coli* inactivation on AgNPs–TiO$_2$ ceramic tiles under low intensity solar simulated light (50 mW/cm$^2$) at different *E. coli* initial concentrations.

### 3.4. Bacterial Inactivation under Indoor Light

Figure 7 shows that the bacterial inactivation can take place under indoor light as well, which further confirms the good photocatalytic ability of such tiles. Under 5 mW/cm$^2$, the photoactive tiles exhibited a 2.5–3 Log reduction of bacterial load. AgNPs–TiO$_2$ tiles may be applied in indoor places as anti-bacterial materials such as in kitchens and bathrooms, as bacteria are more likely to accumulate in indoor places.

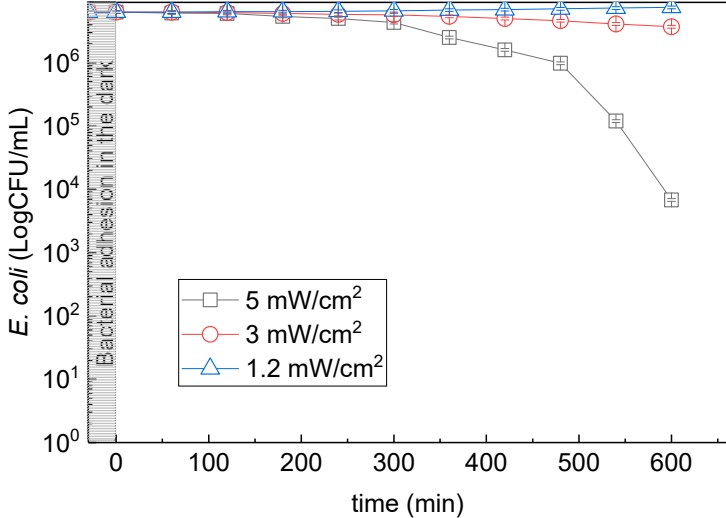

**Figure 7.** *E. coli* inactivation under different intensities of indoor light (see conditions in Section 2.4.4).

### 3.5. Stereomicroscopy and Live/Dead Bacteria at the Interface Of Tiles

Figure 8 depicts stereomicroscopy images captured on the surface of AgNPs–TiO$_2$ tiles at different solar light irradiation times. The green dots represent the alive bacteria, while the red ones are dead bacteria. It can be seen that the green dots completely disappeared within 300 min of irradiation, while red dots appeared instead. The results presented in Figure 5 showed that total bacterial inactivation was reached within 180 min using the plate-count agar method. However, stereomicroscopy results show alive bacteria at this timing. These bacteria are viable non-cultivable as they were not able to develop colonies on the agar.

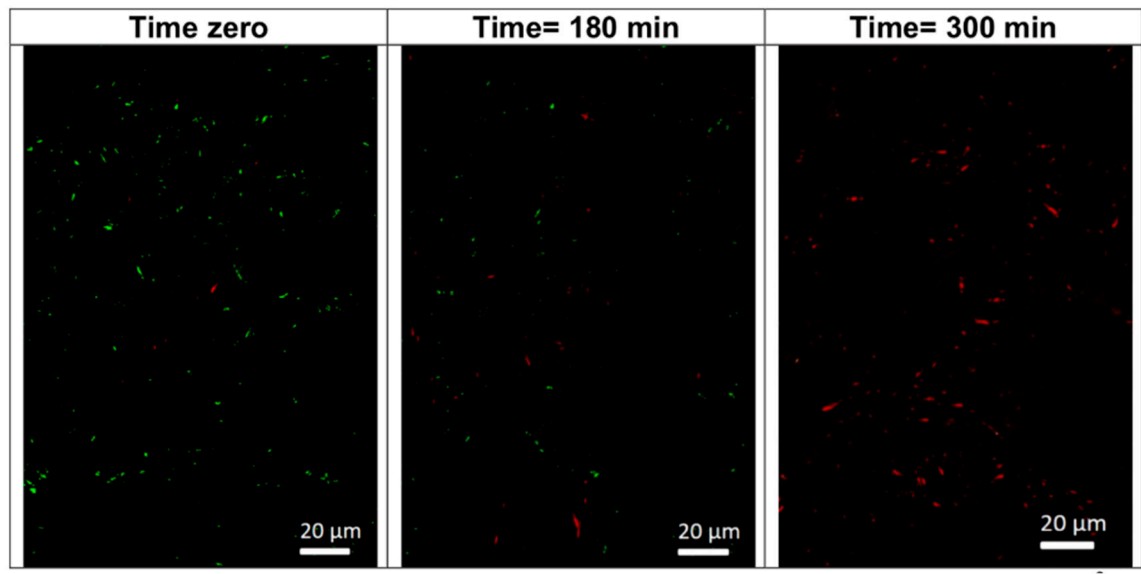

**Figure 8.** *E. coli* inactivation on ceramic tiles under low intensity solar light (50 mW/cm$^2$).

*3.6. Interfacial Potential and Micro-pH at the Tiles' Surface under Low Intensity Solar Simulated Light (50 mW/cm²)*

Figure 9 shows the micro-pH changes during the bacterial inactivation at the interface of the coated tiles. The initial surface potential was around 30 mV. When the light was switched on, the potential shifted to around 50 mV. The interfacial potential then decreased to reach 35 mV. This later phenomenon can be because of the (i) surface stabilization and equilibrium of the possible released ions, and (ii) accumulation of the bacterial biomass on the tile surface. This decrease is concomitant with the bacterial oxidation/destruction, as shown in Figure 5. The surface potential decrease is associated with the increase in cells' permeability, leaching out some ions, especially $K^+$ and $Cl^-$. These changes of the surface potential were preceded by changes in pH. The initial pH of 7.2 decreases to 6.6 within the first 10 min. This can be because of the production of long-lived carboxylic acids intermediates during bacterial degradation. This slight decrease in pH could be also be because of the transformation of OH hydroxyls to $^\bullet$OH. HO are more likely to react with the positive holes to produce $^\bullet$OH than $H_2O$ molecules. Therefore, a slight decrease in pH could be detected during the photocatalytic experiments. Subsequently, the pH recovers its original level when the carboxylic acids decompose, generating $CO_2$. A pH-shift between 6.6 and pH 7.1 takes place afterwards. This is equivalent to a fivefold increase in the concentrations of $H^+$. The short carboxylic acids (branched or not) generated in solution during the bacterial inactivation period present a pKa ~3. The increase of the pH again up to a pH ~7 can be ascribed to the mineralization of the short carboxylic acids into $CO_2$. This is the final step in the bacterial mineralization chain, as reported for many organic compounds and dyes like methylene blue. After total bacterial inactivation (300 min), the interfacial potential and the pH recover to stabilize at 50 mV and 7.1, respectively.

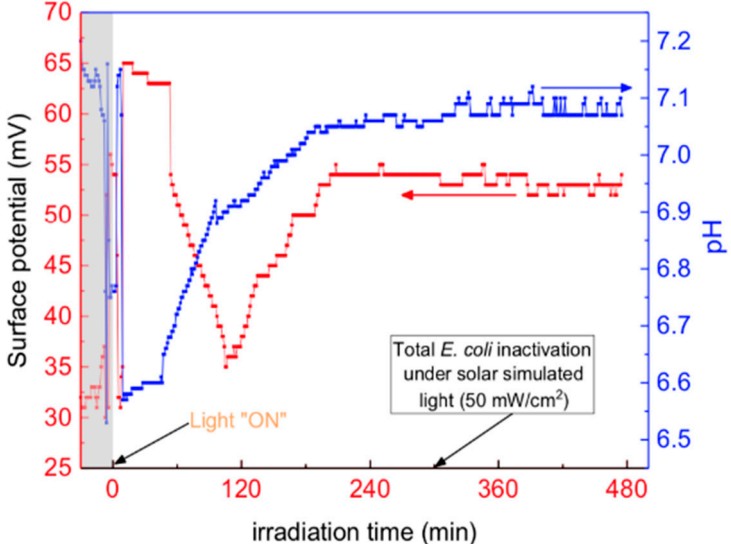

**Figure 9.** Interfacial potential and micro-pH during bacterial inactivation on ceramic tiles under solar simulated light operated at 50 mW/cm².

*3.7. ROS Quenchers*

Figure 10 shows the *E. coli* inactivation on AgNPs–$TiO_2$ ceramic tiles under low intensity solar simulated light (50 mW/cm²) in the presence of $^\bullet$OH radical scavenger (DMSO), photogenerated hole ($h^+$) scavenger (EDTA-2Na), and $O_2^{\bullet-}$ scavenger (SOD). It is readily seen from Figure 10 that the bacterial inactivation kinetics were not seriously affected when $^\bullet$OH radicals were scavenged. This suggests the weak contribution of $^\bullet$OH radical photo-generated at the interface of the AgNPs–$TiO_2$ ceramic tiles. However, the bacterial inactivation kinetics were strongly inhibited when $h^+$ and $O_2^\bullet$ were scavenged.

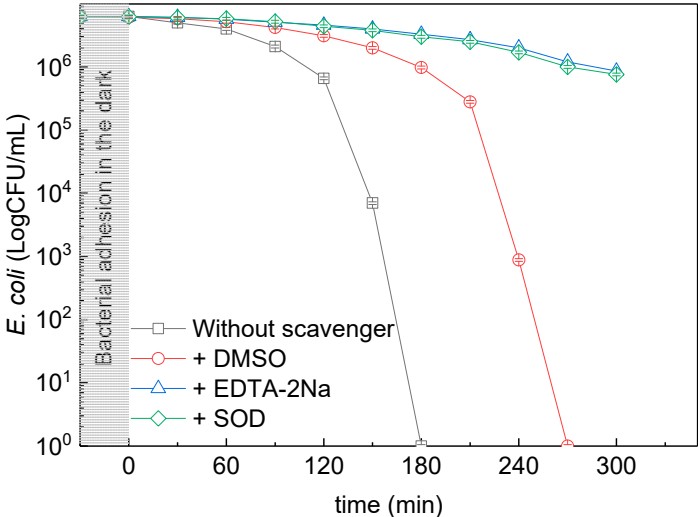

**Figure 10.** *E. coli* inactivation on AgNPs–TiO$_2$ ceramic tiles under low intensity solar simulated light (50 mW/cm$^2$) in the presence of ROS scavengers. DMSO, dimethylsulfoxide; EDTA, ethylenediamine-tetra-acetic acid sodium; SOD, superoxide dismutase.

The generation of O$_2$$^{\bullet-}$ takes place at the conduction band of TiO$_2$ via the reduction of O$_2$ by photogenerated electrons. O$_2$$^{\bullet-}$ is also the source of differnt oxidizing species. The addition of one H$^+$ to O$_2$$^{\bullet-}$ produces HO$_2$$^{\bullet}$. The reduction of O$_2$$^{\bullet-}$ by CB electrons results in the production of H$_2$O$_2$. The oxidation of O$_2$$^{\bullet-}$ by $^{\bullet}$OH, HO$_2$$^{\bullet}$, or positive holes (h$^+$) gives single oxygen species ($^1$O$_2$). Therefore, we may deduce that the scavenging of oxidizing O$_2$$^{\bullet-}$ significantly limits the oxidation of *E. coli* as it is an oxidizing species itself and also the source of many other ROSs. AgNPs deposited onto the TiO$_2$ surface can be a catalytic site to reduce O$_2$ to O$_2$$^{\bullet-}$, as shown in Figure 11. However, some reports suggested that the transfer of electrons from AgNPs to dissolved oxygen is hard and that electrons are most likely to move from excited AgNPs to CB of TiO$_2$. Therefore, owing to the SPR of photoexcited AgNPs, more electrons are transferred from AgNPs to the conduction band of TiO$_2$, which results in higher O$_2$$^{\bullet-}$ generation.

Unlike the photodegradation of organic compounds by ROSs, which is mostly based on the high oxidizing ability of species ($^{\bullet}$OH is the most powerful), for the inactivation of bacteria, the long lifetime of oxidizing species is the most dominant factor rather than its oxidizing potential, because oxidizing species with longer lifetime can penetrate through the outer bacterial membrane and cell wall, resulting in its damage. Coming to this point, H$_2$O$_2$ is the most stable ROS, while the lifetime of O$_2$$^{\bullet-}$ is relatively longer compared with that of other ROSs such as $^{\bullet}$OH. As a result, the powerful oxidant $^{\bullet}$OH was not very effective for the inactivation of *E. coli*, but its presence enhances the speed of *E. coli* inactivation. Some research studies reported that that H$_2$O$_2$ is more efficient for the inactivation of bacteria than $^{\bullet}$OH and $^{\bullet}$O$_2$$^-$ species [29,30]. On the other hand, it is known that one of the advantages of plasmatic photocatalysts doped with noble metal is to increase the lifetime of positive holes for producing more OH$^{\bullet}$ species, herein, we found that the savagening of h$^+$ really decreases the inactivation of *E. coli*, suggesting the high reactivity of h$^+$. It is important to point out that the savaging of h$^+$ not only limits the direct inactivation of *E. coli*, but also mostly suppresses the generation of $^{\bullet}$OH, as this latter are produced via the oxidation of H$_2$O with a photoinduced valence-band hole ($^{\bullet}$OH can be also generated from the reduction of H$_2$O$_2$ via the electrons of CB). Afterwards, the dimerization of $^{\bullet}$OH produces the H$_2$O$_2$ species. Additionally, the H$_2$O$_2$ can also be produced via the double-h$^+$ oxidation of H$_2$O. By analyzing Figure 10, we can notice that the scavenging of one of O$_2$$^{\bullet-}$ or h$^+$ mostly limits the inactivation of *E. coli*, which means that the presence of only O$_2$$^{\bullet-}$ or h$^+$ cannot really oxidize *E. coli*, and a synergetic effect as a result of the combination of O$_2$$^{\bullet-}$ and h$^+$ is a must for efficient *E. coli* inactivation. As mentioned above, the oxidation of O$_2$$^{\bullet-}$ with h$^+$ produces the oxiding species of $^1$O$_2$. However, as $^1$O$_2$ species, an excited state of O$_2$, exhibit a very short lifetime (only 3 μs in

water), the inactivation of *E. coli* by $^1O_2$ is not suggested. The synergistic production of $H_2O_2$ via two pathways, including the reduction of $O_2^{\bullet-}$ and the dimerization of $^{\bullet}OH$, could be the reason behind the fast *E. coli* inactivation along with the effect of other ROSs. We suggest also that the presence of EDTA-2Na or SOD may prevent the reactivity of $H_2O_2$ for *E. coli* inactivation in this system. On the basis of this discussion, the plausible mechanistic pathways for the photocatalytic inactivation of *E. coli* on AgNPs–TiO$_2$ ceramic tiles are shown in Figure 11. Photoproduced ROSs including $^{\bullet}OH$, $O_2^{\bullet-}$, $h^+$, and $H_2O_2$ can directly inactivate *E. coli*. On top of that, AgNPs can play an important photonic and catalytic rule to visible light response of TiO$_2$ and to decrease the recomination of electrons.

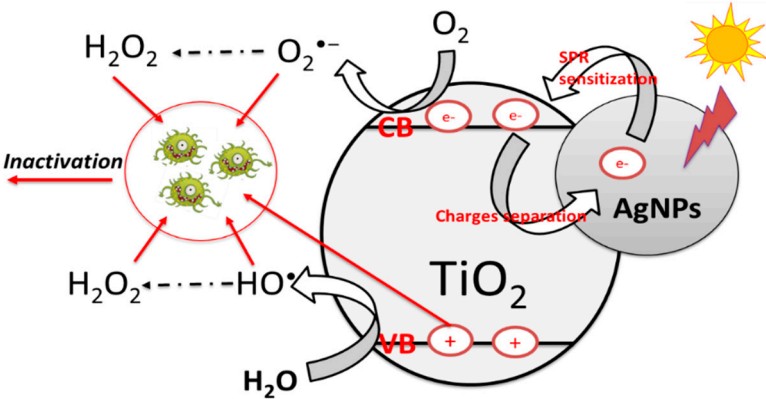

**Figure 11.** Plausible mechanistic pathways for the photocatalytic inactivation of *E. coli* on AgNPs–TiO$_2$ ceramic tiles. SPR, surface plasmon resonance.

The stability of the photocatalytic activity was also investigated. The tile fabrics were washed with water after each photocatalytic test, dried at 80 °C, and a new cycle was started. The photocatalytic activity was seen to be stable over the ten tested cycles. The tiles exhibited similar photocatalytic bacterial inactivation, demonstrating the practical application of these scalable tiles.

## 4. Comparison of the Current Research and a Standard Test According to ISO22196:2011

The results of the present study were finally compared to the antimicrobial activity of the AgNPs–TiO$_2$ photocatalytic ceramic tile and a traditional commercial tile based on the ISO (International Standards Organization) 22196 method, the official standard for measuring the antibacterial efficiency of advanced ceramics [31]. In particular, the main goal was the evaluation of the antibacterial efficiency of the two materials under dark conditions, exclusively exploiting the presence of Ag and not of the photocatalyst.

All specimens (50 × 50 mm$^2$) were rinsed with distilled water and autoclaved at 121 °C for 30 min before the analysis in order to remove any organic residue on the surfaces. For each strain, six tile samples without (controls) and six with photocatalytic coating were investigated for 24 h (time required by the norm), at 37 °C, with an initial inoculated volume of 150 µL *E. coli* (ATCC 8739) and a concentration in the range $10^5$ cells/mL, one/two order of magnitude less than that used in our research in Figure 5. The results are reported in Table 2.

As expected, on the traditional tile, the number of bacteria increased after 24 h owing to the temperature and environment test conditions that facilitate the bacteria growth. The tile doped with AgNPs–TiO$_2$ shows a drastic decrease of the bacteria number with a complete bacteria inactivation after 24 h.

**Table 2.** Data and results of ISO 22196:2011 test.

| Test | Time (h) | UM—Unit Measurements | Results |
|---|---|---|---|
| N. bacteria inoculated on NON photocatalytic tiles ($U_0$) | 0 | Log (cell/cm$^2$) | 4.50 |
| N. bacteria inoculated on photocatalytic tiles ($A_0$) | 0 | Log (cell/cm$^2$) | 4.45 |
| N. bacteria inoculated on NON photocatalytic tiles ($U_t$) | 24 | Log (cell/cm$^2$) | 5.72 |
| N. bacteria inoculated on photocatalytic tiles ($A_t$) | 24 | Log (cell/cm$^2$) | 1.00 |
| Antibacterial activity R * | | Log10 | 4.72 |
| Antibacterial activity R * | | % | 99.99 |

* $R = U_t - A_t$.

## 5. Conclusions

Sustainable plasmonic AgNPs–TiO$_2$ digitally printed onto commercial porcelain-grès tiles showed an excellent continuous photocatalytic antibacterial activity under indoor and outdoor light irradiations. A homogenous surface and a great stability were obtained because of the new economical digital printing synthesis method used for the deposition of AgNPs–TiO$_2$ onto the surface ceramic tiles. A synergistic effect resulting from the combination of charges separation and SPR effect at the interface of AgNPs and TiO$_2$ leads to enhancing the overall photoactivity of the AgNPs–TiO$_2$ for generating more ROS species for bacterial degradation. *E. coli* at a concentration ranging from $10^4$ to $10^7$ CFU/mL was totally inactivated under low intensity solar simulated light (50 mW/cm$^2$). ROS quenching experiments confirmed that the formation of ROSs is the reason behind the bacterial degradation under light irradiation.

Tests in the dark confirmed the role of Ag in the bacteria inactivation even in the absence of light, depending on the number of bacteria present at the ceramic surface. Even at high concentrations of bacteria, the AgNPs–TiO$_2$ tiles are able to eradicate bacteria, hindering biofilm formation.

**Author Contributions:** Conceptualization, C.L.B.; methodology, C.L.B.; validation, C.L.B. and G.C.; formal analysis, C.L.B. and G.C.; investigation: C.L.B., G.C., B.M.B., R.D. and S.R.; data curation, C.L.B. and G.C.; writing—original draft preparation, C.L.B., G.C., B.M.B., R.D., S.R.; writing—review and editing, C.L.B. and G.C.; supervision, C.L.B. All authors have read and agreed to the published version of the manuscript.

**Funding:** This research received no external funding.

**Acknowledgments:** All authors would like to thank Iris Ceramica Group for providing the photocatalytic porcelain-grès tiles samples from the Calacatta SL 300x150 Active 2.0 series.

**Conflicts of Interest:** The authors declare no conflict of interest.

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
