# Peer review of "Digitally Printed AgNPs Doped TiO2 on Commercial Porcelain-Grès Tiles: Synergistic Effects and Continuous Photocatalytic Antibacterial Activity"

_surfaces, doi:10.3390/surfaces3010002_

Round 1

Reviewer 1 Report

Title:

 Digitally Printed AgNPs doped TiO2 ON Commercial  Porcelain-grès Tiles: synergistic effects AND  Continuous Photocatalytic Antibacterial Activity  

Present manuscript discusses the printing of AgNP doped micro-TiO2 on the ceramic tiles and the antibacterials property of the developed surface is studied. Manuscript is written well and provide the information to make the antibacterils surfaces. I would like to recommend the publication of this manuscript after inclusion of following points in the revised manuscript.

Author need to add some work on the coating of photocatalytic materials on various surfaces such as titanium metal plate, Mosquito net, quartz tube, cynospheres and theier techniques in the introduction section. Author can see and cite the ref such as Chemical Engineering Journal 2011, 178, 40-49, Journal of environmental chemical engineering 2016, 4 (1), 319-327, Industrial & Engineering Chemistry Research 2011, 50 (13), 7753-7762, US Patent US 8343282 B2 Author need to provide some more information about digital printing like how it is done what parameter and instruments are required any curing temperature is needed etc. Author need to provide the SEM images of TiO2/AgNP coated and non coate tile surface taken at different magnification to have idea of uniformity of photocatalyst on the surface Author need to carryout the photocatalyst adharance study

Author Response

We would like to thank the reviewer for the pertinent comments. Below, we have responded in detail.

(1) Author need to add some work on the coating of photocatalytic materials on various surfaces such as titanium metal plate, Mosquito net, quartz tube, cynospheres and their techniques in the introduction section. Author can see and cite the ref such as Chemical Engineering Journal 2011, 178, 40-49, Journal of environmental chemical engineering 2016, 4 (1), 319-327, Industrial & Engineering Chemistry Research 2011, 50 (13), 7753-7762, US Patent US 8343282 B2

We are grateful to the reviewer for the comments and citations proposed to add to the paper but the work is specifically related to a coating on porcelain gres tiles for the building sector. They are commercial tiles and therefore it is not appropriate to compare their photocatalytic activity and specifically the antibacterial and coating deposition method, , with other materials.

(2) Author need to provide some more information about digital printing like how it is done what parameter and instruments are required any curing temperature is needed etc.

A new paragraph with reference [5] was added in section 2.2.

(3) Author need to provide the SEM images of TiO2/AgNP coated and non coated tile surface taken at different magnification to have idea of uniformity of photocatalyst on the surface.

A new figure was added as Fig.2.a. Many pictures were already published in other papers so we decided at present to publish only one at 50000X.

(4) Author need to carry out the photocatalyst adherence study

A new sentence was added in section 2.2.

Reviewer 2 Report

Recommendation- major revision 1. introduction part should be revised, 2. All figures revised, 3 explain the working principle of figure 11, How to cal, band gap energy in table 1. 4. Author should be provide full sure spectrum , core level of Ti, O and pure Ag. 5. Author should be provide TEM of Ag particle.

Author Response

We would like to thank the reviewer for the pertinent comments. Below, we have responded in detail.

introduction part should be revised,

Introduction was fully revised and we made some modifications to clarify some parts.

All figures revised,

Figures were revised and a new Figure (Fig.2.a) with an HR-SEM image was added.

Explain the working principle of figure 11,

Figure 11 summarizes the mechanistic pathways of the photocatalytic inactivation of E.Coli (discussion in section 3.7). More sentences were added to the text to explain better and avoid any misunderstanding (section 3.7). 

How to cal, band gap energy in table 1.

Data were elaborated by Kubelka-Moon as mentioned in other papers where all the experimental data were reported.

Author should be provide full sure spectrum, core level of Ti, O and pure Ag.

Ti 2p and O1s spectra are common peaks relative to a normal TiO2 characterized and published several times in literature. No further information can be obtained by them. On the contrary, the Ag 4f profile is very interesting and it is possible to verify the presence of two different kinds of Ag on the surface of our photocatalyst.

Author should be provide TEM of Ag particle. 

We thank the referee for the request but we do not understand the reason as the Ag particles are deposited by impregnation on the TiO2 surface and calcined at 450°C to stabilize them. In this situation, we think it is much more important to characterize the Ag@TiO2 to highlight the Ag dimensions and the uniformity of the particles on the TiO2 than the original Ag nanoparticles.

Reviewer 3 Report

The manuscript includes information obtained from an interesting scientific work, based on the sustainable preparation of Ag-NPs-TiO2 tiles by 3D printing, then these materials were evaluated in the E. coli inactivation, in my opinion this is a complete and good work and the authors have included concluding remarks. However, in order to improve the quality of the manuscript some corrections and complementary information should be included in the revised manuscript. I recommend   accept the manuscript after minor revision. Some recommendations are included as follows:

The ideas included in the paragraph located at Page 2, lines 69-72, should be clarified.

Page 3, lines 99, 112: The information included in this line must be carefully corrected.

The description of the procedure employed for the Ag@TiO2 photoactive tiles should be complemented: What is the use of liquid nitrogen during this procedure? What are the operation parameters employed in the printing processes? What is the commercial reference of the printing machine? What is the composition of the mix used for printing the Ag NPS? What is the concentration of Ag employed in this processes?

The information included in page 3, lines 114 to 124 does not corresponds to materials and methods section.

In section 2.3 the authors have indicated that HR-SEM analyzes were carried out, these results corresponding to SEM images, should be included in the manuscript.

What was the instrument employed for the light intensity measurement?

Section 2.4.5 should be explained in deep.

XRD analysis should be performed.

What it is meaning K+Ag(1%)… in Figure 3? The labels of the samples should be explained in materials and methods section.

Figure 4 should be explained in deep, the information about positions of the XPS peaks and the database employed as the reference should be included in the manuscript.

The authors have indicated that: “These results indicate that the inactivation of E. coli is light-dependent reaction due to the formation of ROS on the surface of AgNPs-TiO2 ”. What is the theoretical foundation, bibliographic reference or scientific explanation of this behavior?. This idea should be clarified and explained in deep.

Why in Figure 5 are only included the results of 8% Ag-NPs-TiO2, the results obtained with other samples should also be included in the manuscript.

Author Response

We would like to thank the reviewer for the pertinent comments. Below, we have responded in detail.

1) The ideas included in the paragraph located at Page 2, lines 69-72, should be clarified.

We have added a sentence in order to clarify the concept related to both the role of Ag species as bactericidal agent and all the possible parameters that can positively influence this action. A couple of references have been added as well (15, 16).

2) Page 3, lines 99, 112: The information included in this line must be carefully corrected.

Line corrected. Thank you

3) The description of the procedure employed for the Ag@TiO2 photoactive tiles should be complemented: What is the use of liquid nitrogen during this procedure? What are the operation parameters employed in the printing processes? What is the commercial reference of the printing machine? What is the composition of the mix used for printing the Ag NPS? What is the concentration of Ag employed in this process?

A new paragraph was added to explain the printing process (section 2.2). In the commercial production, 8% Ag/TiO2 is used.

About the use of liquid nitrogen, a new sentence was added in section 2.2 to clarify the procedure done only to test the washcoat stability.

4) The information included in page 3, lines 114 to 124 does not corresponds to materials and methods section.

All these lines were postponed in section 3.1 - Photocatalyst characterization

5) In section 2.3 the authors have indicated that HR-SEM analyzes were carried out, these results corresponding to SEM images, should be included in the manuscript.

We added a new figure (Fig. 2a).

6) What was the instrument employed for the light intensity measurement?

In the revised manuscript, we have added details about the radiometer used for the light intensity measurement. The added sentence reads, “The light intensity was monitored using a UV radiometer and a global irradiance couple (CUV3 and CM3 Models, Kipp & Zonen)”.

7) Section 2.4.5 should be explained in deep.

In the revised manuscript, we have included the details about the used concentration of the scavengers.

8) XRD analysis should be performed.

We thank the Reviewer for this important remind: as standard procedure, we have run XRD analyses for all the samples, i.e., for both bare-micro titania and Ag-promoted materials and we already published these results elsewhere (moreover, bare titania is a commercial product with well-established characteristics indicated by the producer). However, we have added both a sentence summarizing all the XRD features of the samples object of the present research and one more reference (20), in order to render more complete the description of the materials.

9) What it is meaning K+Ag(1%)… in Figure 3? The labels of the samples should be explained in materials and methods section.

A new line was added in the experimental section 2.2 to clarify the presence of three different samples in Fig.3.

10) Figure 4 should be explained in deep, the information about positions of the XPS peaks and the database employed as the reference should be included in the manuscript.

The reference due to the attribution of the single Ag peaks is already present (ref 23). On the contrary, a new sentence (and reference) was added to explain the presence of the large peak at 377 eV, as the Kronos K1077 TiO2 has contamination by K in its original formulation.

11) The authors have indicated that: “These results indicate that the inactivation of E. coli is light-dependent reaction due to the formation of ROS on the surface of AgNPs-TiO2 ”. What is the theoretical foundation, bibliographic reference or scientific explanation of this behavior?. This idea should be clarified and explained in deep.

We thank the reviewer for drawing our attention to this point. In the revised version of the manuscript, we have added the theoretical foundation behind this observation. The corrected paragraph in page 7 reads: “However, in the presence of AgNPs-TiO2 tiles under light, a significant inactivation of E. coli that increases as a function of light intensity increase. This justifies the semiconductor behaviour of the Ag-species at the interface with bacteria [New References 27, 28]. Thus, these results indicate that the inactivation of E. coli is light-dependent reaction due to the formation of ROS on the surface of AgNPs-TiO2.”

12) Why in Figure 5 are only included the results of 8% Ag-NPs-TiO2, the results obtained with other samples should also be included in the manuscript.

8% Ag-NPs-TiO2 is the sample chosen for industrial production because it guarantees the best photocatalytic performance. One sentence was now added in section 2.2.